# CARE: Confidence-Aware Ratio Estimation for Medical Biomarkers

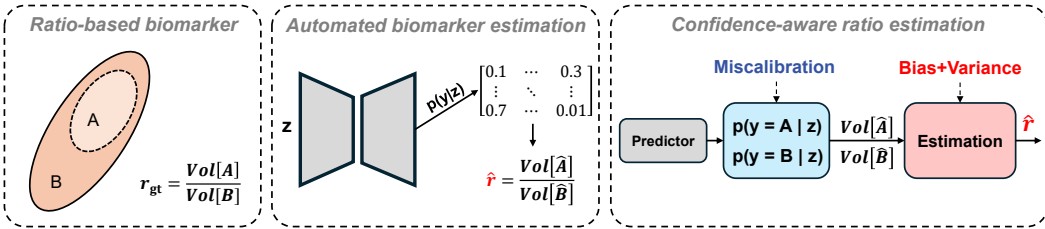

Figure 1: **Overview of CARE.** In automated medical imaging analysis, biomarkers are often computed from network predictions. To quantify the uncertainty of ratio-based biomarkers, we introduce CARE, a confidence-aware estimation method that provides reliable confidence intervals.

## ABSTRACT

Ratio-based biomarkers – such as the proportion of necrotic tissue within a tumor – are widely used in clinical practice to support diagnosis, prognosis, and treatment planning. These biomarkers are typically estimated from soft segmentation outputs by computing region-wise ratios. Despite the high-stakes nature of clinical decision making, existing methods provide only point estimates, offering no measure of uncertainty. In this work, we propose a unified *confidence-aware* framework for estimating ratio-based biomarkers. Our uncertainty analysis stems from two observations: i) the probability ratio estimator inherently admits a statistical confidence interval regarding local randomness (bias and variance), ii) the segmentation network is not perfectly calibrated. We conduct a systematic analysis of error propagation in the segmentation-to-biomarker pipeline and identify model miscalibration as the dominant source of uncertainty. We leverage tunable parameters to control the confidence level of the derived bounds, allowing adaptation towards clinical practice. Extensive experiments show that our method produces statistically sound confidence intervals, with tunable confidence levels, enabling more trustworthy application of predictive biomarkers in clinical workflows.

## 1 INTRODUCTION

The success of deep learning in medical image analysis, particularly since the introduction of UNet architectures (Ronneberger et al., 2015; Isensee et al., 2021), has enabled automated segmentation of anatomical and pathological structures across a range of clinical imaging tasks. However, segmentation is rarely the end goal in clinical workflows. Instead, it often serves as an intermediate step toward computing biomarkers – quantitative metrics such as volumes (Popordanoska et al., 2021; Rousseau et al., 2025; Kazerouni et al., 2023; Abdusalomov et al., 2023) and fraction scores (Ronneberger et al., 2015; Isensee et al., 2021; Bahna et al., 2022; Kim et al., 2008) – that are used to assess disease progression, guide treatment decisions, or monitor therapeutic response. As shown in Fig. 1, the ratio-based biomarker is usually derived from two volume measurements. Since segmentation models provide per-pixel prediction, they allow automated ratio estimation. Nevertheless, relying solely on a single point estimate offers no quantification of uncertainty, which limits the clinical adoption and undermines its value as a decision-making reference. To address this, we study confidence-aware ratio estimation for medical biomarkers.

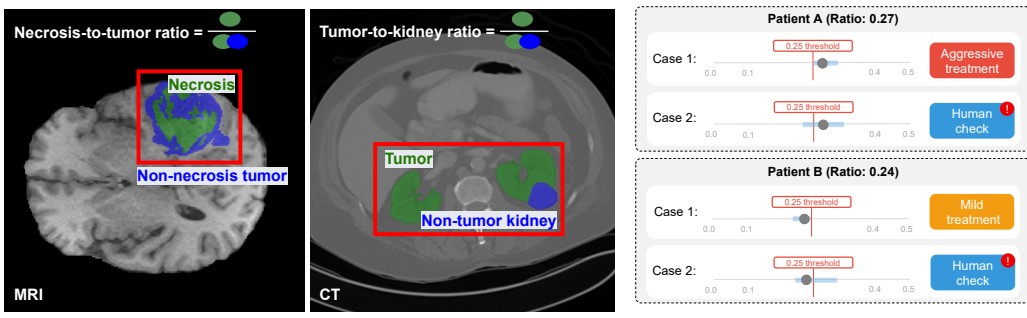

(a) Biomarkers                                   (b) Clinical support

Figure 2: **Medical background of ratio estimation and its role in clinical support. (a)**: Ratio-based biomarkers (Baid et al., 2021; Myronenko et al., 2023) exist in many organs and modalities. **(b)**: An illustrative example where a high-risk threshold is defined as 0.25; CARE calls for human check when the confidence interval crosses the threshold.

As shown in Fig. 2a, ratio-based biomarkers are widely utilized across various organs and imaging modalities. These biomarkers provide quantitative measures for clinical decision-making. For example, the necrosis-to-tumor ratio (NTR) (Henker et al., 2019; 2017) quantifies the proportion of necrotic (non-viable) tissue within a tumor, while the tumor-to-kidney ratio (TKR) Herts et al. (2002) indicates the extent of tumor infiltration within the kidney. Accurate estimation of these biomarkers is crucial for supporting personalized treatments and monitoring their efficacy. A straightforward method for computing these ratios involves using segmentation models to identify the subregion and the whole foreground region, and then calculating the ratio based on averaged softmax confidence scores over these regions. However, the interpretation of this point estimate can change once the confidence interval is considered. As illustrated in Fig. 2b, consider a clinical threshold of 0.25 for initiating aggressive treatment. Based on point estimates alone, Patient A would receive aggressive treatment (high ratio) while Patient B would receive mild treatment (low ratio). However, if the associated confidence interval spans the decision threshold (case 2), the estimation is flagged for mandatory expert review to mitigate potential misdiagnosis risk. Such double-check procedures are essential in clinical practice, as they provide an additional safeguard for patients and enhance the robustness of downstream decision-making. Despite their clinical importance, most efforts still focus on improving upstream segmentation accuracy (Ronneberger et al., 2015; Isensee et al., 2021; Hatamizadeh et al., 2021), while the uncertainty and reliability of downstream ratio-based biomarkers remain largely unexplored.

We propose CARE, the *first confidence-aware estimation framework specifically for ratio-based biomarkers*, offering several key advantages: i) **guaranteed coverage**, *i.e.*, the actual coverage probability of containing the true ratio is greater than the stated nominal confidence level; ii) instance-wise **adaptiveness**, *i.e.*, providing dynamic intervals that capture varying uncertainty degrees; iii) **tunable** confidence level with user-controlled tightness; iv) applicable as a **plug-in** module to any pretrained NN requiring neither architectural modifications nor training from scratch; v) computationally **efficient**, avoiding multiple sampling or repeated forward passes.

Furthermore, we identify sources of error and quantify their individual impacts on the overall confidence intervals. Specifically, we establish a ratio estimator bound using Markov's inequality (Resnick, 2003) and derive a squared error estimator from volume predictions. We also address the issue of overconfident predictions from deep learning models, which represents another critical limitation undermining ratio estimation reliability. To quantify the error caused by miscalibration, we provide theoretical insights into the relationship between model calibration and ratio estimation and propose a miscalibration-based bound, building on recent advances in calibration error (CE) estimation (Guo et al., 2017; Popordanoska et al., 2022).

Beyond theoretical contributions, our framework is designed for practical clinical deployment. CARE operates with linear time complexity $\mathcal{O}(n)$, making it lightweight and reliable. Compared with Bayesian methods, CARE is well-suited for clinical settings where real-time performance is essential and large-scale computing resources are unavailable. Experiments further confirm that the proposed confidence bounds are conservative, adaptive and computationally efficient.

In summary, our main **contributions** are:

① We propose CARE, a principled framework for confidence-aware ratio estimation for medical biomarkers in an automated estimation workflow (Sec. 3).

② We analyze the sources of error across the entire segmentation-to-biomarker pipeline (Sec. 3) and empirically demonstrate that miscalibration is the dominant factor (Sec. 4).

③ Experiments confirm that CARE effectively tracks the prediction uncertainty, represented as the coverage of erroneous predictions and the distinguishability of segmentation difficulties (Sec. 4).

## 2 RELATED WORK

**Ratio-based biomarkers** are quantitative metrics that express the relative size, volume, or intensity of a target anatomical structure as a proportion of a reference region (Fig. 1). They are widely used across clinical domains to capture compositional, structural and functional changes, enabling standardized assessment of disease progression and treatment response. Examples include: ejection fraction – representing the fraction of blood ejected from the ventricle during each cardiac cycle; coronary artery stenosis – quantifying the percent narrowing of a coronary vessel, and fat fraction – measuring the proportion of fat within an organ such as liver or kidney. Ratio-based biomarkers are particularly valuable for detailed tumor characterization (Fig. 2). Key metrics include necrosis-to-tumor ratio (NTR) and core-to-tumor ratio (CTR), which quantify the internal structure of the tumor, as well as tumor invasion rate, which reflects the extent of tumor infiltration into surrounding tissues. In summary, the ratio-based measures offer standardized, comparable metrics that can be applied across imaging modalities, organs, and disease contexts.

Typically, clinicians compute these ratios using volumetric information from imaging data (*e.g.*, MRI) (Henker et al., 2019; 2017). With the advancement of computational pathology and the growing availability of annotated medical data, recent studies (Ye et al., 2023) have developed AI-based workflows for automated ratio assessment. These methods offer scalable and consistent evaluations, effectively overcoming the limitations of subjective human judgment in manual assessments. Despite promising developments, existing methods typically provide only point estimates (Ho et al., 2020), neglecting the associated uncertainty. Although intuitive, results computed from the outputs of segmentation networks inherit the known overconfidence tendency of neural networks (Guo et al., 2017). As a result, naïve ratio estimations from miscalibrated outputs are often biased from true values. Current research predominantly focuses on improving network calibration and segmentation accuracy (Rousseau et al., 2025; Wang et al., 2023; Mehrtash et al., 2020; Wang et al., 2022; Hatamizadeh et al., 2021), while overlooking the downstream task of biomarker estimation. Our work addresses this gap by proposing a confidence-aware framework for ratio estimation from segmentation models.

**Uncertainty quantification (UQ)** provides many statistical methods to estimate prediction uncertainty. *Conformal prediction (CP)* (Vovk et al., 1999; Papadopoulos et al., 2002; Vovk et al., 2005; Angelopoulos & Bates, 2021; Karimi & Samavi, 2023; Angelopoulos & Bates, 2021) constructs prediction intervals that guarantee valid coverage under finite samples, without any distributional assumptions. Its key strength is the distribution-free nature and finite-sample validity, providing strong theoretical guarantees regardless of the base predictive model. *Resampling methods* are non-parametric techniques for estimating the sampling distribution of a statistic, applicable when the underlying distribution is unknown or difficult to derive. Specifically, *Bootstrapping* (Mooney et al., 1993; Freedman, 1981) repeatedly samples $N$ data points with replacement from the original data, whereas *subsampling* (Politis & Romano, 1994) takes a subset of the original data without replacement, repeating the process multiple times to construct an empirical distribution of the statistic. *Bayesian methods* achieve robust segmentation by averaging multiple predictions, using techniques like deep ensemble (Lakshminarayanan et al., 2017) and Monte Carlo dropout (Srivastava et al., 2014). These approaches enable confidence interval estimation by computing standard deviation across several feedforward inferences. However, they require proper prior specification and cannot provide tunable quantiles due to the limited number of inference samples (usually $\leq 10$). Moreover, these universal methods are either computationally expensive or fail to provide informative conclusions.

**Calibration error (CE)** estimation has attracted extensive research attention (Kull & Flach, 2015; Vaicenavicius et al., 2019; Kumar et al., 2019; Zhang et al., 2020; Popordanoska et al., 2022; Gruber & Buettner, 2022). In medical segmentation, classwise and canonical calibration error are used to

evaluate per-structure and overall calibration levels. Derived from individual channel masks, the classwise CE in multi-class segmentation simplifies to binary CE for each channel. In addition, Popordanoska et al. (2021) proves that the absolute value of volume bias (V-Bias) is upper-bounded by CE. Many calibration methods like temperature scaling (Guo et al., 2017) and isotonic regression (Zadrozny & Elkan, 2002) have been proposed to improve the calibration of classification scores. However, no previous work analyzes how miscalibration affects downstream ratio-based estimates.

## 3 METHODS

We begin with relevant definitions in Sec. 3.1 to establish the theoretical background. In Sec. 3.2, we present our main contribution, the uncertainty decomposition and corresponding confidence intervals.

### 3.1 PRELIMINARIES

The ratio-based biomarker is clinically defined as the ratio between two volumes $V_A$ and $V_B$ (Henker et al., 2019; 2017). We consider the ratio estimation within a standard segmentation framework, where $V_A$ and $V_B$ are calculated from predicted probabilities.

**Definition 3.1** (Ratio from Segmentation Networks)**.** Given per-pixel inputs $\{z_i\}_{i=1}^n$, labels $\{y_{A,i}, y_{B,i}\}_{i=1}^n$ and segmentation model $g\colon z_i \to g_A(z_i), g_B(z_i) \in [0,1]$, the labeled ratio $r_{\text{gt}}$ and predicted ratio $\hat{r}$ within $n$ pixels are calculated by:

$$r_{\text{gt}} = \frac{\bar{y_A}}{\bar{y_B}} = \frac{\sum_{i=1}^n y_{A,i}}{\sum_{i=1}^n y_{B,i}}, \text{ and } \hat{r} = \frac{\bar{g_A}}{\bar{g_B}} = \frac{\sum_{i=1}^n g_A(z_i)}{\sum_{i=1}^n g_B(z_i)}. \tag{1}$$

**Proposition 3.2** (Conformal Prediction for Regression (Shafer & Vovk, 2008))**.** *Given groundtruth* $r_{gt}$*, prediction* $\hat{r}$ *and the absolute error residual* $\text{AE}_{\text{r}} \coloneqq |r_{gt} - \hat{r}|$*, let* $q_{r,\delta}$ *denote the* $1 - \delta$ *quantile of the instance-wise* $\text{AE}_{\text{r}}$ *on a validation (calibration) set. Then, with probability at least* $1 - \delta$

$$r_{\text{gt}} \in [\hat{r} - q_{r,\delta}, \hat{r} + q_{r,\delta}], \tag{2}$$

From Def. 3.1, the predicted ratio $\hat{r}$ is determined by the probability volumes predicted by the network. Since the network is not perfectly calibrated, quantifying the uncertainty in its predictions is closely tied to assessing the uncertainty of the derived biomarker. To this end, we introduce two metrics: volume bias and the calibration error.

**Definition 3.3** (Volume Bias (Popordanoska et al., 2021))**.** Given a segmentation model $g\colon \mathcal{Z} \to [0,1]$ that predicts the probability of $y \in \{0,1\}$, the volume bias is defined as:

$$\text{V-Bias}(g) \coloneqq \mathbb{E}_{(z,y)\sim P}\left[g(z) - y\right]. \tag{3}$$

**Definition 3.4** (Calibration Error (Kumar et al., 2019))**.** Given a model $g\colon \mathcal{Z} \to [0,1]$ that predicts the probability of $y \in \{0,1\}$, the calibration error is defined as:

$$\text{CE}(g) \coloneqq \mathbb{E}_{(z,y)\sim P}\left[|g(z) - \mathbb{E}\left[y = 1 \mid g(z)\right]|\right], \tag{4}$$

**Proposition 3.5** (The Relationship of V-Bias and CE (Popordanoska et al., 2021))**.** *Given segmentation model* $g\colon \mathcal{Z} \to [0,1]$*, the absolute value of volume bias is upper bound by the calibration error, i.e.,* $|\text{V-Bias}(g)| \leq \text{CE}(g)$*.*

Leveraging the upper bound relationship and statistical properties, we decompose the uncertainty of the ratio estimation pipeline and give respective confidence intervals in Sec. 3.2.

### 3.2 CARE: CONFIDENCE-AWARE RATIO ESTIMATION

In this section, we illustrate our insight of uncertainty analysis based on two key observations, as shown in Fig. 3. The first observation is that the ratio estimator $\hat{r} = \frac{\bar{y}}{\bar{x}}$ is subject to instance-wise randomness, which we capture using statistical tools such as Markov's inequality to derive an *estimation-based interval*. The second observation is that the network is not perfectly calibrated, introducing a global, model-level error affecting both the numerator and denominator; this gives rise to the *calibration-based interval*. Combining these two sources yields the overall uncertainty bound.

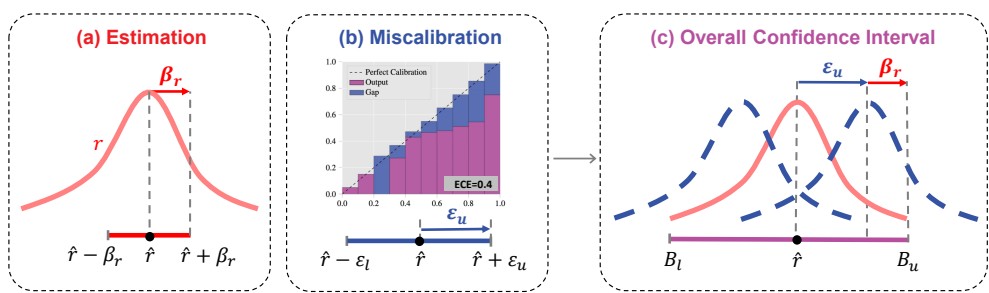

Figure 3: **Our confidence interval considering estimation and miscalibration.** **(a)** shows Markov bounds from the estimator. **(b)** illustrates the prediction offset $\epsilon_{l,u}$ due to miscalibration. **(c)** is the overall confidence interval $r \in [B_l, B_u]$.

**Estimation-based interval.** Van Kempen & Van Vliet (2000) provides an approximated theoretical result for ratio statistics. However, their derivation critically relies on the assumption that the addends in $\bar{x}$ and $\bar{y}$ are independent. Therefore, the result in Van Kempen & Van Vliet (2000) is not directly applicable in imaging analysis for violating spatial patterns. As a remedy, we construct Markov bounds as an estimation-based confidence interval for $\hat{r}$ using Markov inequality (Resnick, 2003). Although this approach leads to more conservative bounds, it avoids strong assumptions such as pixel independence, making it more applicable to image data.

**Proposition 3.6** (Estimation-based Confidence Interval). *Given an estimator $\hat{r} = \frac{\bar{y}}{\bar{x}}$ of the fraction $r = \frac{\mathbb{E}[y]}{\mathbb{E}[x]}$ with random variables $x$ and $y$, it holds with at least $1 - \alpha$ probability that*

$$r \in [\hat{r} - \beta_{r,\alpha},\ \hat{r} + \beta_{r,\alpha}], \tag{5}$$

*where $\beta_{r,\alpha} := \frac{\sqrt{\text{SE}_{\hat{r}}}}{\sqrt{\alpha}}$ is the half-width of the bound, and $\text{SE}_{\hat{r}} := \mathbb{E}\left[(\hat{r} - r)^2\right]$ is the expected squared error.*

Then we conduct a Taylor expansion of $\text{SE}_{\hat{r}}$ to receive an approximation we can estimate in practice.

**Proposition 3.7.** *Assume all central moments of the independently and identically distributed random variables $(x_1, y_1), \ldots, (x_n, y_n) \sim \mathbb{P}_{xy}$ in the estimator $\hat{r} = \frac{\bar{y}}{\bar{x}}$ exist, then we have*

$$\text{SE}_{\hat{r}} = \frac{1}{n}\left(\frac{\text{Var}(y)}{\mu_x} + \text{Var}(x)\frac{\mu_y^2}{\mu_x^4} - 2\,\text{Cov}(x,y)\frac{\mu_y}{\mu_x^3}\right) + O\left(\frac{1}{n^2}\right). \tag{6}$$

The proof is given in the appendix. Then the estimator is:

$$\widehat{\text{SE}_{\hat{r}}} := \frac{1}{n}\left(\frac{\hat{\sigma}_y^2}{\bar{x}} + \frac{\hat{\sigma}_x^2 \bar{y}^2}{\bar{x}^4} - 2\frac{\hat{\sigma}_{xy}\bar{y}}{\bar{x}^3}\right), \tag{7}$$

with the sample variances $\hat{\sigma}_x^2 = \frac{1}{n-1}\sum_i (x_i - \bar{x})^2$, $\hat{\sigma}_y^2 = \frac{1}{n-1}\sum_i (y_i - \bar{y})^2$, and sample covariance $\hat{\sigma}_{xy} = \frac{1}{n-1}\sum_i (x_i - \bar{x})(y_i - \bar{y})$. Under i.i.d. assumption, the estimator $\widehat{\text{SE}_{\hat{r}}}$ is consistent, i.e., $\widehat{\text{SE}_{\hat{r}}} \to \text{SE}_{\hat{r}}$ in probability for $n \to \infty$. The proof is presented in the appendix B.1.

**Calibration-based interval.** The estimation-based bounds involve local uncertainty that stems from statistical properties. Then we analyze the second source of uncertainty: volume bias caused by miscalibration. Inspired by Prop. 3.2, we propose a fine-grained calibration-based confidence interval by considering the uncertainty of target (A) and RoI (B) regions separately. Unlike vanilla conformal prediction, where analysis starts from the final $\hat{r}$, we adopt quantiles of $V_A$ and $V_B$ and report the corresponding interval as CARE (V-Bias). As described in Proposition 3.5, V-Bias of target (A) and RoI (B) regions is upper bounded by their calibration errors, *i.e.*, $|\text{V-Bias}(g_A)| \leq \text{CE}(g_A)$, $|\text{V-Bias}(g_B)| \leq \text{CE}(g_B)$. This motivates the more conservative interval named as CARE (ECE).

**Proposition 3.8** (Calibration-based Confidence Interval). *Consider a segmentation model $g(z) = (g_A(z), g_B(z))$ with the random variable $z$ representing pixel inputs of instance $I$, and targets $y_A$*

and $y_B$. On a validation (calibration) set $\mathcal{D}_{cal}$, define $q_{A,\delta/2}$ and $q_{B,\delta/2}$ as the $1 - \delta/2$ quantile of the instance-wise volume bias or calibration errors of $g_A$ and $g_B$. Then, it holds with at least $1 - \delta$ probability that

$$\frac{\mathbb{E}\left[y_A \mid I\right]}{\mathbb{E}\left[y_B \mid I\right]} \in \left[\frac{\mathbb{E}\left[g_A(z) \mid I\right]}{\mathbb{E}\left[g_B(z) \mid I\right]} - \epsilon_{l,\delta}, \frac{\mathbb{E}\left[g_A(z) \mid I\right]}{\mathbb{E}\left[g_B(z) \mid I\right]} + \epsilon_{u,\delta}\right], \tag{8}$$

where $\epsilon_{l,\delta} := \frac{\mathbb{E}[g_A(z)]}{\mathbb{E}[g_B(z)]} - \frac{\mathbb{E}[g_A(z)] - q_{A,\delta/2}}{\mathbb{E}[g_B(z)] + q_{B,\delta/2}}$, $\epsilon_{u,\delta} := \frac{\mathbb{E}[g_A(z)] + q_{A,\delta/2}}{\mathbb{E}[g_B(z)] - q_{B,\delta/2}} - \frac{\mathbb{E}[g_A(z)]}{\mathbb{E}[g_B(z)]}$ are the widths of the lower and upper calibration bounds, respectively.

The proof is presented in the appendix B.2. In experiments, CARE (V-Bias) takes the quantile of |V-Bias| as $q_{N,T}$ while CARE (ECE) considers ECE (Guo et al., 2017) quantiles. To combine both intervals, we make the following statement, which is analogous to multiple testing. This way, we can consider both uncertainties in practice.

**Proposition 3.9** (Overall Confidence Interval). *Assume we have a ratio estimator $\hat{r} = \frac{\sum_i g_A(z_{i,I})}{\sum_i g_B(z_{i,I})}$ for pixel measurements $\{z_{i,I}\}_{i=1}^n$ of an instance $I$ based on neural network outputs $g(z_{i,I}) = (g_A(z_{i,I}), g_B(z_{i,I}))$. Let $y_A$ and $y_B$ be the instance-wise target random variables. Then, it holds with at least $1 - \alpha - \delta$ probability that*

$$\frac{\mathbb{E}\left[y_A \mid I\right]}{\mathbb{E}\left[y_B \mid I\right]} \in \left[\frac{\sum_i g_A\left(z_{i,I}\right)}{\sum_i g_B\left(z_{i,I}\right)} - \epsilon_{l,\delta} - \beta_{r,\alpha}, \frac{\sum_i g_A\left(z_{i,I}\right)}{\sum_i g_B\left(z_{i,I}\right)} + \epsilon_{u,\delta} + \beta_{r,\alpha}\right], \tag{9}$$

*where $\beta_{r,\alpha}$ is defined as in Prop. 3.6 and $\epsilon_{l,\delta}, \epsilon_{u,\delta}$ as in Prop. 3.8.*

The interval width $w = B_u - B_l$ measures the uncertainty level, as a result, a wide interval over thresholds alarms for manual examination. In the experiments, we alternate through various $\alpha$ and $\delta$ for a fixed $\alpha + \delta$ with grid search to observe the impact on the interval width. This way, we can choose the smallest interval under a desired coverage rate.

# 4 EXPERIMENTS

## 4.1 EXPERIMENTAL SETUP

**Datasets.** We evaluate on two brain tumor segmentation datasets: MSD-Task01 (Antonelli et al., 2022) and BraTS21 (Baid et al., 2021), which include 484 and 1251 MRI volumes, respectively, with four modalities (T1, T2, T1ce, FLAIR) and four annotations (edema, necrosis, enhancing tumor, background). Their necrosis-to-tumor ratio (NTR) is defined as the ratio of necrosis $V_N$ to the whole tumor area $V_T$ (edema, necrosis, enhancing). In addition, we include KiTS23 (Myronenko et al., 2023), a CT dataset of 489 kidney volumes. Its tumor-to-kidney ratio (TKR) is defined as $\frac{V_{tumor}}{V_{whole\ kidney}}$. A nested five-fold cross-validation is used for all datasets. In the outer loop, four folds are used for training and validation, and the remaining one fold for testing. Within the inner loop, 10% of the training data is held out as a validation set $\mathcal{D}_{cal}$ to estimate the quantile of V-Bias and ECE. Predicted ratio $\hat{r}$ and labeled ratio $r_{gt}$ are calculated from Def. 3.1.

**Segmentation models.** We conduct experiments using nnUNet (Isensee et al., 2021), nnFormer (Zhou et al., 2021) and UNETR++ (Zhou et al., 2021). All models are trained using four-modality MRI scans, label-based supervision and softmax activation under a single A100 GPU. We investigate different loss functions as ensemble baselines: cross-entropy (XE) (Bishop & Nasrabadi, 2006), soft Dice (SD) (Milletari et al., 2016), TopK (Deng et al., 2009), and their combinations (XE-SD, Top10-SD), while use XE-SD for the main results.

**UQ baselines.** To control the confidence level to be $C = 0.68$, we adopt a quantile for each baseline method. For conformal prediction (Vovk et al., 1999; Papadopoulos et al., 2002) and CARE, we obtain quantiles from the validation set. Specifically, for conformal prediction we take the 0.68 quantile $(0.68Q)$ of $AE_r$ from the validation set as the half-width (Prop. 3.2), while for CARE we adopt dynamic ECE quantiles or V-Bias quantiles by conducting a grid search under the constraint of $1 - \alpha - \beta = 0.68$ (Prop. 3.9). To implement resampling, we repeatedly sample pixels from an instance and calculate its ratio estimate for 100 times, then adopt the $[0.16Q, 0.84Q]$ from 100 repetitions as the 0.68 confidence level. For a volume of $N$ pixels, we take $0.1N$ random pixels each

Table 1: **Comparison of the coverage guarantee on MSD-Task01 dataset** ($C = 0.68$). We report the overall coverage rate (%) on test-set. CARE always satisfies the desired confidence level, while other methods fall below in most cases.

| Coverage (%) | nnUNet$_{2d}$ | nnUNet$_{3d}$ | nnFormer | UNETR++ |
|---|---|---|---|---|
| **Subsampling** | $6.19_{\pm0.77}$ | $9.28_{\pm0.92}$ | $5.74_{\pm0.72}$ | $8.22_{\pm0.91}$ |
| **Bootstrap** | $5.34_{\pm0.61}$ | $8.18_{\pm0.62}$ | $5.53_{\pm0.75}$ | $8.12_{\pm0.71}$ |
| **Conformal prediction** | $71.34_{\pm2.00}$ | $67.01_{\pm3.57}$ | $67.39_{\pm1.66}$ | $65.75_{\pm2.16}$ |
| **CARE (V-Bias)** | $93.61_{\pm1.14}$ | $86.60_{\pm1.49}$ | $81.92_{\pm1.31}$ | $76.43_{\pm2.21}$ |
| **CARE (ECE)** | $94.22_{\pm0.99}$ | $93.61_{\pm0.71}$ | $87.94_{\pm0.97}$ | $89.58_{\pm1.02}$ |

time without replacement for subsampling (Politis & Romano, 1994), and sample $N$ pixels with replacement each time for bootstrapping (Mooney et al., 1993). For Bayesian methods, conducting numerous forward passes to estimate a "tunable" quantile is computationally impractical; thus, we report the results of three standard deviations ($3\sigma$).

**Metrics.** We evaluate the performance of various methods across four criteria: i) *Coverage guarantee*: ability to achieve the desired confidence level, quantified by coverage rate, ii) *Adaptiveness*: capacity to capture sample variability (*e.g.*, prediction error) and segmentation difficulty (*e.g.*, tumor size); iii) *Tunability*: flexibility to choose a user-specified confidence level; iv) *Practical deployment*: whether the method operates as a plug-in module (non-intrusive to the model architecture) and maintains computational efficiency (without requiring multiple sampling or repeated inference steps).

## 4.2 RESULTS

We demonstrate the claimed properties in Sec. 1 of our method: coverage guarantee, adaptiveness, tunability and practical deployment characteristics (plug-in compatibility and computational efficiency). Moreover, we analyze the uncertainty source to get an insight into the dominant component.

**Coverage guarantee.** As described in Sec.1, a conservative confidence interval achieves coverage probability higher than the nominal confidence level, *i.e.*, achieving over 68% coverage when aiming for 68% confidence level. We report coverage rate (%) of different UQ methods at 0.68 confidence level in Table 1, which measures *the proportion of samples whose true values fall within the confidence intervals*. Empirically, our intervals show higher likelihoods of satisfying the prescribed confidence level of 0.68 compared with other UQ methods. Considering the suboptimal performance of sampling-based methods, our following comparison focuses on CP and CARE.

**Adaptiveness.** Beyond achieving the guaranteed coverage rate, the confidence interval should be sample-adaptive to identify unreliable predictions effectively. We demonstrate this capability by examining the "dataset-level interval" distribution of MSD-Task01 in Fig. 4. As observed, CP values lie within a narrow range and thus fail to effectively indicate which samples are unreliable. In contrast, our method produces intervals that vary significantly in width. Given an interval width threshold, our method can effectively trigger alarms for cases with wide intervals (indicating high uncertainty), thereby overcoming CP's limitation of producing uniformly narrow confidence ranges.

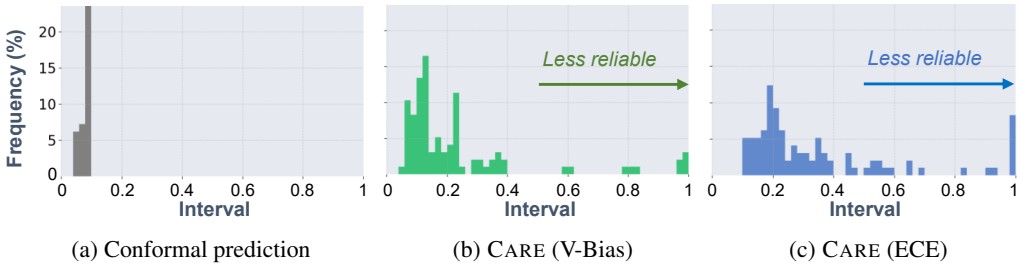

        (a) Conformal prediction        (b) CARE (V-Bias)        (c) CARE (ECE)

Figure 4: **Comparison of interval distribution on MSD-Task01 dataset** ($C = 0.68$). We report the frequency histogram of NTR intervals in test-set, where CARE triggers a human-check alarm when the interval is too wide.

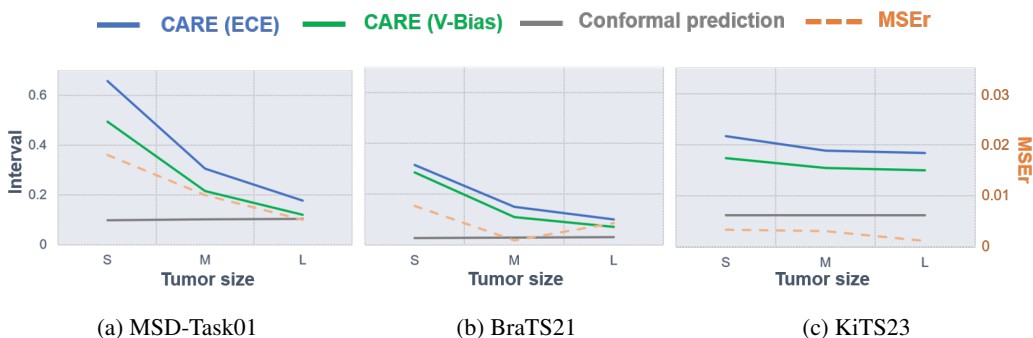

(a) MSD-Task01       (b) BraTS21       (c) KiTS23

Figure 5: **Comparison of tumor-size adaptiveness on nnUNet$_{3d}$** ($C = 0.68$). For each dataset, we report the average interval width in three groups categorized by tumor sizes. Intuitively, interval width should reflect the MSE$_r$ tendency in ---. Compared with the indistinguishable results of CP, CARE becomes wider for small tumors (hard samples) and tighter for large ones (simple samples).

Furthermore, the uncertainty should correlate appropriately with segmentation difficulty. For instance, small tumors are hard to detect and segment for their small size, low contrast and susceptibility to noise. Empirically, hard samples with small sizes or blurry boundaries tend to yield erroneous predictions (large mean squared error), necessitating wider intervals to ensure coverage. To validate this adaptive behavior, we present fine-grained analysis of MSE$_r$ (error measures) and interval width (uncertainty measures) in Fig. 5. Specifically, we report NTR for Fig. 5a and 5b, and report TKR for Fig. 5c. We stratify tumors into small (S), medium (M), and large (L) categories based on the $\frac{1}{3}$ and $\frac{2}{3}$ quantiles of tumor sizes in test-set. As illustrated, our intervals widths are proportional to the segmentation difficulty: smaller, more challenging tumors receive wider intervals, while larger, easier-to-segment tumors receive narrower intervals.

**Tunability.** CARE offers two variants that allow clinicians to select either conservative or informative bounds by choosing CARE (ECE) or (V-Bias). To demonstrate tunability and coverage guarantee across different confidence levels, we report coverage rates for varying confidence thresholds on two biomarkers: NTR and CTR in Fig. 6. The coverage rate is expected to increase proportionally with the increased confidence level. However, conformal prediction shows significant limitations: it only achieves adequate coverage at isolated points ($C = 0.7$ for NTR and $C = 0.6, 0.7$ for CTR), while falling below the target confidence level at all other tested thresholds. Additionally, conformal prediction consistently fails to provide adequate coverage for small tumors (NTR-S) across the entire confidence range, as demonstrated in Fig. 6c. In contrast, both variants of our method consistently achieve coverage rates above the desired confidence level.

**Other baselines.** Bayesian UQ methods like deep ensemble (Lakshminarayanan et al., 2017) and Monte Carlo dropout (Srivastava et al., 2014) require modifications to model architectures or training procedures, in contrast to the previously discussed plug-in methods. For practical usage, CP and

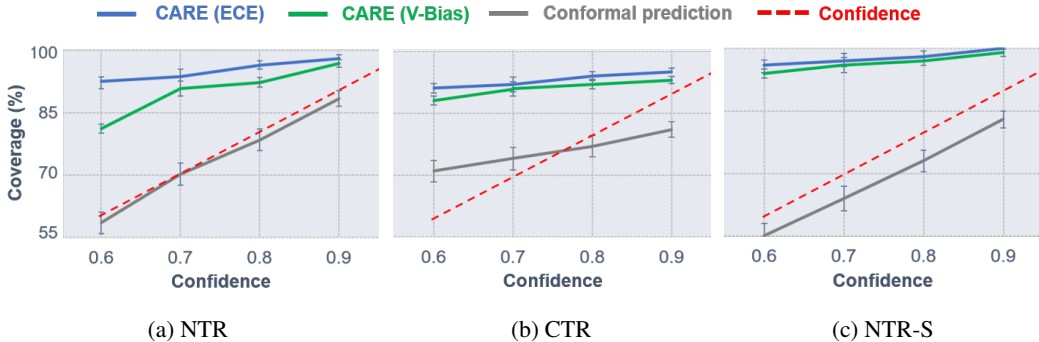

(a) NTR       (b) CTR       (c) NTR-S

Figure 6: **Performance comparison across confidence levels on MSD-Task01 and nnUNet$_{3d}$.** Methods over --- achieve the desired coverage. CARE is tunable to maintain the target coverage at any confidence level and outperforms CP in detecting high-risk small tumors (NTR-S).

CARE achieve linear computational complexity with minimal computational overhead. In comparison, Bayesian methods face significant practical limitations in clinical settings. Performing numerous ensemble predictions or dropout inferences is computationally expensive and often impractical for real-time applications.

In Table 2, we implement two ensemble configurations on BraTS21 using nnUNet$_{3d}$: i) training with five random seeds, and ii) training with five different loss functions (XE, SD, Top10, XE-SD, Top10-SD). For dropout configurations, we add dropout layers to the nnUNet decoder with dropout probabilities of 0.3 and 0.5, and perform 20 stochastic forward passes. The interval widths of Bayesian methods are set as $3\sigma$ to provide a very conservative interval. Nevertheless, we find that the intervals of Bayesian methods are still too narrow to provide valid intervals, most likely due to the lack of an appropriate prior (Noci et al., 2021).

**Uncertainty source analysis.** As described in Sec. 3.2, we decompose uncertainty into miscalibration and intrinsic bias of ratio estimation. We validate this empirically by analyzing 10 randomly selected volumes from BraTS21, calculating ECE-based ($I_{\text{ECE}}$) and overall CARE confidence intervals ($I$). The results in Fig. 7 show that $I_{\text{ECE}}$ dominates the overall interval $I$, indicating that model miscalibration is the primary uncertainty source in ratio estimation.

Table 2: **Comparison of UQ methods on BraTS21 with nnUNet$_{3d}$.** Ensemble and dropout methods provide too narrow NTR intervals.

|  | Interval | Coverage (%) |
|---|---|---|
| Ensemble$_{\text{loss}}$ | $0.088_{\pm 0.003}$ | $46.03_{\pm 1.21}$ |
| Ensemble$_{\text{seed}}$ | $0.041_{\pm 0.002}$ | $43.03_{\pm 1.13}$ |
| Dropout$_{0.3}$ | $0.033_{\pm 0.001}$ | $27.09_{\pm 1.02}$ |
| Dropout$_{0.5}$ | $0.038_{\pm 0.001}$ | $29.63_{\pm 1.03}$ |

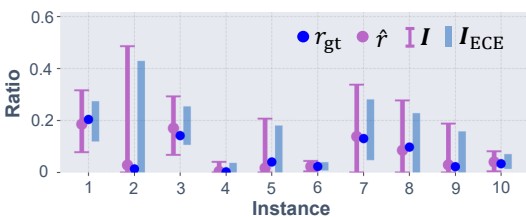

Figure 7: **Uncertainty decomposition of 10 samples**. Miscalibration is the main contributor to the overall uncertainty.

**Discussion about post-hoc calibration** We calculate ECE and our intervals on different temperatures in appendix A.2. For overconfident models, temperature$_{>1}$ helps for better calibrated performance. When ECE decreases, our interval becomes tighter consistently.

## 5 CONCLUSION

We propose CARE, a confidence-aware framework for estimating ratio-based biomarkers from segmentation network outputs. Our method addresses a common limitation of prior works that focus solely on point estimates without confidence guarantees. We disentangle two key sources of uncertainty, *i.e.* network prediction error and statistical bias. Our empirical findings highlight that miscalibration is a dominant contributor to uncertainty. Our framework offers several practical advantages: it operates as a model-agnostic plugin module, provides sample-level adaptive uncertainty estimates in a single forward pass without requiring multiple sampling, and allows users to flexibly adjust confidence levels. In summary, this work represents an important step toward trustworthy deployment of deep learning in clinical settings by providing practitioners with both accurate biomarker estimates and reliable confidence bounds.

**Limitations and future work.** Despite the practical advantages, our work has several limitations. First, we assume that the validation and test sets are drawn from the same distribution. Although it is standard in supervised learning settings, but may not hold under domain shifts. In practice, domain shifts arise due to differences in scanners, acquisition protocols, or patient populations. As a result, our confidence interval may not remain valid in these scenarios. Addressing this challenge with label-free calibration error estimators (e.g. Wang et al. (2020); Popordanoska et al. (2024)) is a promising direction for future work. Second, the calibration quality of the underlying segmentation network has an impact on the tightness of the derived confidence intervals. Specifically, when the calibration error is large, the resulting confidence intervals may become overly conservative. Improving calibration in segmentation networks would directly translate into narrower, more informative confidence intervals within our approach. Finally, while our framework shows good performance on public datasets, clinical validation is needed to assess its real-world impact on decision-making and patient outcomes.

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

# Appendix

In Appendix A, we present additional experimental results, relevant to our methodology and in support of the main paper. In Appendix B, we offer the proofs of propositions in the main paper. Finally, we claim the LLM usage in Appendix C.

## A MORE EMPIRICAL RESULTS

This section presents additional empirical results. To further examine different components of our framework, we first provide more visualization in Sec. A.1 to further demonstrate the coverage guarantee and adaptiveness of our confidence intervals. In the main paper, we observe that miscalibration is the main cause of uncertainty. As an extension, we conduct post-hoc calibration in Sec. A.2 to report ECE and our confidence interval under different temperatures. Finally, in Sec. A.3, we replace the default ECE metric (Guo et al., 2017) with KDE-based calibration error (Popordanoska et al., 2022), highlighting the flexibility of our framework with respect to calibration error estimators. All experiments are conducted at the 0.68 confidence level.

### A.1 COVERAGE GUARANTEE AND ADAPTIVENESS

In the main paper, we just visualize the confidence intervals of 10 randomly selected samples in Fig. 7. To provide a more comprehensive, "bird-eye" view of our method's behavior, we extend this analysis to the whole test samples in Fig. A, where we plot $r_{gt}$ and the confidence intervals $I$ under three methods. For clarity, the sample indices are omitted. As shown here, the ground-truth ratio $r_{gt}$ (blue point) always lies well within our predicted confidence interval, while for conformal prediction, $r_{gt}$ occasionally falls outside the interval when the upper or lower bounds are too narrow. The conservative property of CARE is particularly important in clinical settings to provide a reliable and informative reference.

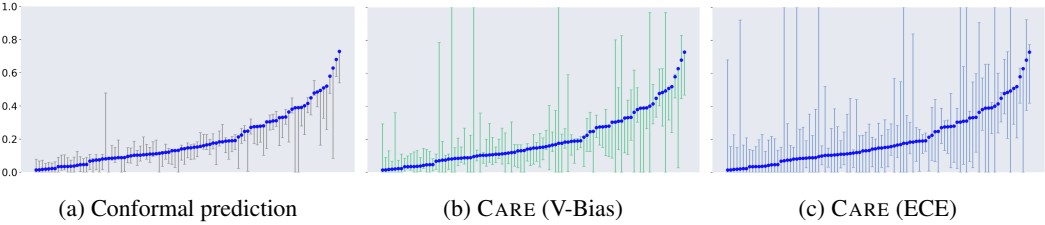

| (a) Conformal prediction | (b) CARE (V-Bias) | (c) CARE (ECE) |

Figure A: **Visualization of our confidence intervals on MSD and nnUNet$_{3d}$**. The x-axis represents all test samples sorted by labeled ratio $r_{gt}$, and the y-axis displays the valid range of ratio estimates. As an extension of Fig. 5, we show our adaptiveness towards tumor sizes by three different pretrained models in Fig. B. As the previous setting, all samples are divided into three groups: small (S), medium (M) and large (L). Since CARE is a plug-in module with the model-agnostic nature, the adaptiveness holds on all pretrained models.

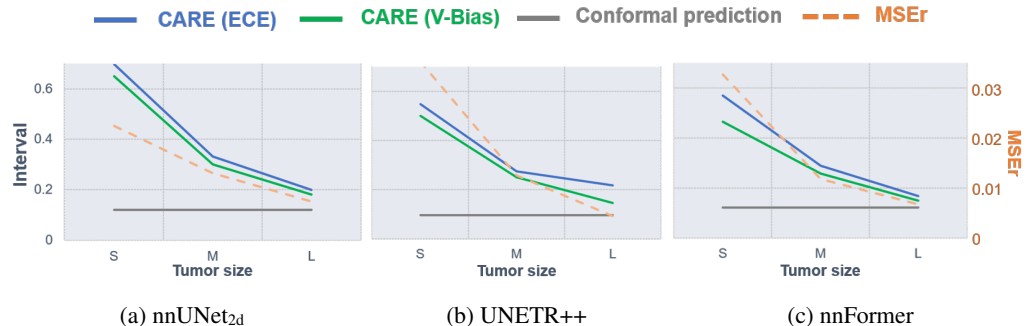

| (a) nnUNet$_{2d}$ | (b) UNETR++ | (c) nnFormer |

Figure B: **Comparison of tumor-size adaptiveness on MSD.** CARE is adaptive to different tumor sizes across all models.

## A.2 POST-HOC CALIBRATION

Our main contribution is the confidence interval for ratio estimation. Nevertheless, we acknowledge that the calibration property is very important for downstream tasks, as demonstrated by Fig. 7. In this section, we report different temperature parameters on a pretrained nnUNet$_{3d}$, to observe the effect of post-hoc calibration. We report the ECE of necrosis and tumor (ECE$_{N, T}$) and average interval width of CARE (ECE) under different temperatures in Table A. For illustration, we scale up ECE by 100. As observed, both ECE and our interval width decrease as the temperature increases. The consistency demonstrates that better-calibrated models have tighter confidence intervals.

Table A: **Comparison of different temperature parameters on MSD and nnUNet$_{3d}$.** When the temperature moves towards better calibration (ECE ↓), our interval becomes narrower (Interval ↓).

| Temperature | ECE$_N$ | ECE$_T$ | CARE (ECE) |
|:---:|:---:|:---:|:---:|
| 0.1 | $0.112_{\pm0.004}$ | $0.166_{\pm0.006}$ | $0.405_{\pm0.023}$ |
| 0.5 | $0.101_{\pm0.005}$ | $0.141_{\pm0.004}$ | $0.364_{\pm0.012}$ |
| 1.0 | $0.098_{\pm0.006}$ | $0.134_{\pm0.003}$ | $0.355_{\pm0.025}$ |
| 1.5 | $0.097_{\pm0.004}$ | $0.131_{\pm0.006}$ | $0.353_{\pm0.027}$ |
| 2.0 | $0.097_{\pm0.002}$ | $0.129_{\pm0.004}$ | $0.353_{\pm0.023}$ |
| 3.0 | $0.096_{\pm0.029}$ | $0.128_{\pm0.005}$ | $0.352_{\pm0.026}$ |
| 4.0 | $0.096_{\pm0.003}$ | $0.127_{\pm0.007}$ | $0.351_{\pm0.024}$ |
| 8.0 | $0.095_{\pm0.008}$ | $0.126_{\pm0.004}$ | $0.349_{\pm0.035}$ |

## A.3 OTHER CALIBRATION ERROR ESTIMATORS

We replace the calibration error estimator ECE (Guo et al., 2017) with ECE$_{kde}$ (Popordanoska et al., 2022) for comparison. In segmentation task, we use binary calibration error which corresponds to Beta kernel in ECE$_{kde}$. Since kernel computation is much expensive than bins, and all pixels together will cause OOM error, we sample $10^4$ pixels once for ECE$_{kde}$ estimation, and repeat 5 times to report the average value. Observed from 10 volumes in Fig. C, ECE$_{kde}$ tends to provide wider bounds, which is suitable for conservative preference. Notably, the estimator is flexible to plug into our framework. We don't aim to give any recommendations, depending on the priority of tightness or informativeness. For a conservative estimator, the alarm thresholds of the interval width should also increase to avoid over-checking. As future work, these differentiable ECE estimator may facilitate obtaining tighter confidence intervals through carefully designed optimization..

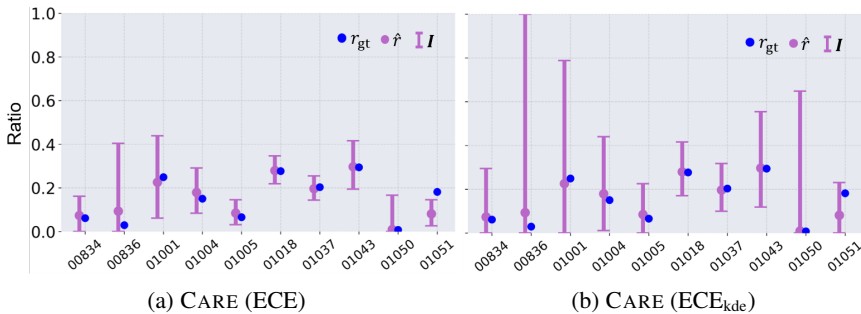

(a) CARE (ECE)                    (b) CARE (ECE$_{kde}$)

Figure C: **Comparison of different calibration error estimators on BraTS21 and nnUNet$_{3d}$.** Adopting (a) ECE (Guo et al., 2017) is generally tighter and more informative than (b) KDE.

# B PROOFS

In this section, we give the corresponding proof of Markov bounds (B.1) and miscalibration bounds (B.2) mentioned in Sec. 3.2 of the main paper. In addition, we derive a debiased estimator in Sec. B.3.

## B.1 Markov Bounds

Van Kempen & Van Vliet (2000) provides a confidence interval of the ratio estimator $\frac{\bar{y}}{\bar{x}}$ based on asymptotic normal assumptions and by using the variance $\sigma_r^2 := \text{Var}\left(\frac{\bar{y}}{\bar{x}}\right)$. However, adopting their results assumes that all pixels are independently and identically distributed, i.e., $(x_1, y_1), \ldots, (x_n, y_n) \overset{\text{i.i.d.}}{\sim} \mathbb{P}_{xy}$. In addition, they perform multiple approximation steps, and some approximations happen within the square operator. How the estimator behaves facing a violation of these assumptions is unknown in practice. In the following, we prove the alternative approach, we proposed in the main paper, which is based on Markov's inequality (Resnick, 2003). For conciseness, the "$\approx$" sign is avoided while we directly note the remainder terms for a rigorous analysis.

To avoid relying on any distribution assumptions, we construct a confidence interval via Markov's inequality for the estimator $\hat{r} = \frac{\bar{y}}{\bar{x}}$ and target $r = \frac{\mu_y}{\mu_x}$. We have

$$\mathbb{P}\left(|\hat{r} - r| \geq k\sqrt{\text{SE}_{\hat{r}}}\right) = \mathbb{P}\left((\hat{r} - r)^2 \geq k^2 \text{SE}_{\hat{r}}\right) \leq \frac{1}{k^2} \tag{10}$$

with the squared error $\text{SE}_{\hat{r}} := \mathbb{E}\left[(\hat{r} - r)^2\right]$. We emphasize that in general $\sqrt{\text{SE}_{\hat{r}}} \neq \sigma_r$.

In main paper, we denote $\alpha := \frac{1}{k^2}$ as the non-coverage probability. For instance, adopting the $1 - \alpha = 75\%$ confidence interval corresponds to $\alpha = \frac{1}{k^2} = 0.25$ or $k = 2$. Then the half-width of confidence interval is $2\sqrt{\text{SE}_{\hat{r}}}$, *i.e.*, two times the root squared error. This is more conservative than using the normal assumption, but requires no distribution assumption.

Now, we compute the squared error via Taylor expansion (Spivak, 2006). First, note that

$$\text{SE}_{\hat{r}} = \mathbb{E}\left[\left(\frac{\bar{y}}{\bar{x}} - \frac{\mu_y}{\mu_x}\right)^2\right] = \mathbb{E}\left[\frac{\bar{y}^2}{\bar{x}^2}\right] - 2\frac{\mu_y}{\mu_x}\mathbb{E}\left[\frac{\bar{y}}{\bar{x}}\right] + \frac{\mu_y^2}{\mu_x^2}. \tag{11}$$

We perform a Taylor expansion of $\frac{\bar{y}^2}{\bar{x}^2}$ around $\frac{\mu_y}{\mu_x}$ to compute its expectation:

$$\begin{aligned}
\frac{\bar{y}^2}{\bar{x}^2} &= \frac{\mu_y^2}{\mu_x^2} + 2\left(\bar{y} - \mu_y\right)\frac{\mu_y}{\mu_x^2} - 2\left(\bar{x} - \mu_x\right)\frac{\mu_y^2}{\mu_x^3} \\
&\quad + \left(\bar{y} - \mu_y\right)^2 \frac{1}{\mu_y} + 3\left(\bar{x} - \mu_x\right)^2 \frac{\mu_y^2}{\mu_x^4} - 4\left(\bar{y} - \mu_y\right)\left(\bar{x} - \mu_x\right)\frac{\mu_y}{\mu_x^3} \\
&\quad + \sum_{i,j:\; i+j \geq 3} \left(\bar{y} - \mu_y\right)^i \left(\bar{x} - \mu_x\right)^j \frac{\partial^{i+j}}{\partial^i \mu_y \partial^j \mu_x} \frac{\mu_y^2}{\mu_x^2}
\end{aligned} \tag{12}$$

from which follows

$$\begin{aligned}
\mathbb{E}\left[\frac{\bar{y}^2}{\bar{x}^2}\right] &= \frac{\mu_y^2}{\mu_x^2} + \frac{\text{Var}\left(\bar{y}\right)}{\mu_y} + 3\,\text{Var}\left(\bar{x}\right)\frac{\mu_y^2}{\mu_x^4} - 4\,\text{Cov}\left(\bar{x}, \bar{y}\right)\frac{\mu_y}{\mu_x^3} \\
&\quad + \sum_{i,j:\; i+j \geq 3} \mathbb{E}\left[\left(\bar{x} - \mu_x\right)^i \left(\bar{y} - \mu_y\right)^j\right] \frac{\partial^{i+j}}{\left(\partial \mu_x\right)^i \left(\partial \mu_y\right)^j} \frac{\mu_y^2}{\mu_x^2}.
\end{aligned} \tag{13}$$

Assuming $(x_1, y_1), \ldots, (x_n, y_n) \sim \mathbb{P}_{xy}$ are i.i.d. further simplifies the terms, like in the following. Markov's inequality does not require this assumption, so a violation does not invalidate our approach. Then, it holds that

$$\text{Var}\left(\bar{x}\right) = \frac{1}{n}\text{Var}\left(x\right), \quad \text{Var}\left(\bar{y}\right) = \frac{1}{n}\text{Var}\left(y\right), \quad \text{Cov}\left(\bar{x}, \bar{y}\right) = \frac{1}{n}\text{Cov}\left(x, y\right). \tag{14}$$

Further, for all $a = 1, \ldots n$ let $z_{k,a} = x_a$ and $\mu_{z_k} = \mu_x$ if $1 \leq k \leq i$, and $z_{k,a} = y_a$ and $\mu_{z_k} = \mu_y$ if $i < k \leq m := i + j$. Then

$$
\begin{aligned}
&\mathbb{E}\left[ (\bar{x} - \mu_x)^i (\bar{y} - \mu_y)^j \right] \\
&= \frac{1}{n^{i+j}} \mathbb{E}\left[ \left( \sum_{a=1}^n x_a - \mu_x \right)^i \left( \sum_{a=1}^n y_a - \mu_y \right)^j \right] \\
&= \frac{1}{n^m} \mathbb{E}\left[ \prod_{k=1}^m \left( \sum_{a=1}^n z_{k,a} - \mu_{z_k} \right) \right] \\
&= \frac{1}{n^m} \sum_{l=1}^m \sum_{a_l=1}^n \mathbb{E}\left[ \prod_{k=1}^m (z_{k,a_k} - \mu_{z_k}) \right]
\end{aligned}
\tag{15}
$$

For all $a_k$ holds that $\mathbb{E}\left[ \prod_{k=1}^m (z_{k,a_k} - \mu_{z_k}) \right] = 0$ if there exists any non-duplicate index value, due to independence. It follows that we can reduce the number of indices by at least half, which reduces the number of addends by a polynomial:

$$
\begin{aligned}
&\frac{1}{n^m} \underbrace{\sum_{l=1}^m \sum_{a_l=1}^n \mathbb{E}\left[ \prod_{k=1}^m (z_{k,a_k} - \mu_{z_k}) \right]}_{n^m \text{ addends}} \\
&= \frac{1}{n^m} \underbrace{\sum_{l=1}^{\lfloor m/2 \rfloor} \sum_{a_l=1}^n \mathbb{E}\left[ \prod_{k=1}^m (z_{k,a_k} - \mu_{z_k}) \right]}_{n^{\lfloor m/2 \rfloor} \text{ addends}} \\
&= \frac{1}{n^{\lceil m/2 \rceil}} \underbrace{\frac{1}{n^{\lfloor m/2 \rfloor}} \sum_{l=1}^{\lfloor m/2 \rfloor} \sum_{a_l=1}^n \mathbb{E}\left[ \prod_{k=1}^m (z_{k,a_k} - \mu_{z_k}) \right]}_{=:C_{ij}}.
\end{aligned}
\tag{16}
$$

Note that $C_{ij} \in [-B_m, B_m]$ with $B_m := \max_{\{i,j=0,\ldots m \mid i+j \leq m\}} \left| \mathbb{E}\left[ (x - \mu_x)^i (y - \mu_y)^j \right] \right|$, therefore, the convergence rate depends not only on the data size $n$ but also on how the moments grow with $m$.

Using Eqn. 14 and Eqn. 16 gives

$$
\begin{aligned}
\mathbb{E}\left[ \frac{\bar{y}^2}{\bar{x}^2} \right] &= \frac{\mu_y^2}{\mu_x^2} + \frac{\operatorname{Var}(y)}{n\mu_y} + 3\operatorname{Var}(x) \frac{\mu_y^2}{n\mu_x^4} - 4\operatorname{Cov}(x,y) \frac{\mu_y}{n\mu_x^3} \\
&\quad + \sum_{i,j:\ i+j \geq 3} \frac{1}{n^{\lceil (i+j)/2 \rceil}} C_{ij} \frac{\partial^{i+j}}{(\partial \mu_x)^i (\partial \mu_y)^j} \frac{\mu_y^2}{\mu_x^2}.
\end{aligned}
\tag{17}
$$

Similarly, we use Taylor expansion for $\frac{\bar{y}}{\bar{x}}$ around $\frac{\mu_y}{\mu_x}$ to get

$$
\begin{aligned}
\frac{\bar{y}}{\bar{x}} &= \frac{\mu_y}{\mu_x} + (\bar{y} - \mu_y) \frac{1}{\mu_x} - (\bar{x} - \mu_x) \frac{\mu_y}{\mu_x^2} \\
&\quad + 0 + (\bar{x} - \mu_x)^2 \frac{\mu_y}{\mu_x^3} - (\bar{y} - \mu_y)(\bar{x} - \mu_x) \frac{1}{\mu_x^2} \\
&\quad + \sum_{i,j:\ i+j \geq 3} (\bar{y} - \mu_y)^i (\bar{x} - \mu_x)^j \frac{\partial^{i+j}}{(\partial \mu_x)^i (\partial \mu_y)^j} \frac{\mu_y}{\mu_x},
\end{aligned}
\tag{18}
$$

which results in

$$
\begin{aligned}
\frac{\mu_y}{\mu_x} \mathbb{E}\left[\frac{\bar{y}}{\bar{x}}\right] &= \frac{\mu_y^2}{\mu_x^2} + \operatorname{Var}(\bar{x}) \frac{\mu_y^2}{\mu_x^4} - \operatorname{Cov}(\bar{y}, \bar{x}) \frac{\mu_y}{\mu_x^3} \\
&\quad + \sum_{i,j:\ i+j\geq 3} \mathbb{E}\left[(\bar{y}-\mu_y)^i (\bar{x}-\mu_x)^j\right] \frac{\mu_y}{\mu_x} \frac{\partial^{i+j}}{(\partial\mu_x)^i (\partial\mu_y)^j} \frac{\mu_y}{\mu_x} \\
&= \frac{\mu_y^2}{\mu_x^2} + \operatorname{Var}(x) \frac{\mu_y^2}{n\mu_x^4} - \operatorname{Cov}(x,y) \frac{\mu_y}{n\mu_x^3} \\
&\quad + \sum_{i,j:\ i+j\geq 3} \frac{1}{n^{\lceil (i+j)/2 \rceil}} C_{ij} \frac{\mu_y}{\mu_x} \frac{\partial^{i+j}}{(\partial\mu_x)^i (\partial\mu_y)^j} \frac{\mu_y}{\mu_x}.
\end{aligned}
\tag{19}
$$

Inserting Eqn. 17 and Eqn. 19 into Eqn. 11 results in

$$
\begin{aligned}
\mathrm{SE}_{\hat{r}} &= 2\frac{\mu_y^2}{\mu_x^2} + \frac{\operatorname{Var}(y)}{n\mu_x} + 3\operatorname{Var}(x)\frac{\mu_y^2}{n\mu_x^4} - 4\operatorname{Cov}(x,y)\frac{\mu_y}{n\mu_x^3} \\
&\quad + \sum_{i,j:\ i+j\geq 3} \frac{1}{n^{\lceil (i+j)/2\rceil}} C_{ij} \frac{\partial^{i+j}}{(\partial\mu_x)^i(\partial\mu_y)^j}\frac{\mu_y^2}{\mu_x^2} \\
&\quad - 2\Big(\frac{\mu_y^2}{\mu_x^2} + \operatorname{Var}(x)\frac{\mu_y^2}{n\mu_x^4} - \operatorname{Cov}(x,y)\frac{\mu_y}{n\mu_x^3} \\
&\quad + \sum_{i,j:\ i+j\geq 3} \frac{1}{n^{\lceil (i+j)/2\rceil}} C_{ij} \frac{\mu_y}{\mu_x}\frac{\partial^{i+j}}{(\partial\mu_x)^i(\partial\mu_y)^j}\frac{\mu_y}{\mu_x}\Big) \\
&= \frac{1}{n}\left(\frac{\operatorname{Var}(y)}{\mu_x} + \operatorname{Var}(x)\frac{\mu_y^2}{\mu_x^4} - 2\operatorname{Cov}(x,y)\frac{\mu_y}{\mu_x^3}\right) \\
&\quad + \underbrace{\sum_{i,j:\ i+j\geq 3} \frac{1}{n^{\lceil (i+j)/2\rceil}} C_{ij}\left(\frac{\partial^{i+j}}{(\partial\mu_x)^i(\partial\mu_y)^j}\frac{\mu_y^2}{\mu_x^2} - \frac{2\mu_y}{\mu_x}\frac{\partial^{i+j}}{(\partial\mu_x)^i(\partial\mu_y)^j}\frac{\mu_y}{\mu_x}\right)}_{\in O\left(\frac{1}{n^2}\right)}.
\end{aligned}
\tag{20}
$$

Consequently, we may estimate $\mathrm{SE}_{\hat{r}}$ via

$$
\widehat{\mathrm{SE}}_{\hat{r}} := \frac{1}{n}\left(\frac{\hat{\mu}_y \hat{\sigma}_x^2}{\hat{\mu}_x^4} + \frac{\hat{\sigma}_y^2}{\hat{\mu}_x} - 2\frac{\hat{\mu}_y \hat{\sigma}_{xy}}{\hat{\mu}_x^3}\right),
\tag{21}
$$

which is consistent since the estimators $\hat{\mu}_y = \frac{1}{n}\sum_i y_i$, $\hat{\mu}_x = \frac{1}{n}\sum_i x_i$, $\hat{\sigma}_y^2 = \frac{1}{n-1}\sum_i (y_i - \hat{\mu}_y)^2$, $\hat{\sigma}_x^2 = \frac{1}{n-1}\sum_i (x_i - \hat{\mu}_x)^2$, and $\hat{\sigma}_{xy} = \frac{1}{n-1}\sum_i (x_i - \hat{\mu}_x)(y_i - \hat{\mu}_y)$ are consistent as well.

### B.2 VOLUME TO RATIO CONFIDENCE INTERVALS

Note that if $a \notin [b,c] \subseteq \mathbb{R}_{>0}$ then $\frac{1}{a} \notin \left[\frac{1}{c}, \frac{1}{b}\right]$ since $x \mapsto \frac{1}{x}$ is strictly negative monotone. We also make use of the subadditivity of probability measures (Resnick, 2003) given by

$$
\mathbb{P}\left(\bigcup_i A_i\right) \leq \sum_i \mathbb{P}(A_i).
\tag{22}
$$

This is also known as Boole's inequality. Denote the random variable $z$ representing pixel inputs of image instance $I$. Let $q_{T,\alpha}$ and $q_{N,\alpha}$ be empirically determined on a validation set as the $1-\alpha$

quantile of the image-wise calibration errors for $g_T$ and $g_N$. Then, for $\alpha \in [0, 1]$ it holds

$$
\begin{aligned}
\alpha &= \frac{\alpha}{2} + \frac{\alpha}{2} \\
&\geq \mathbb{P}\left(\mathrm{CE}_{N,I} \geq q_{N,\alpha/2}\right) + \mathbb{P}\left(\mathrm{CE}_{T,I} \geq q_{T,\alpha/2}\right) \\
&\geq \mathbb{P}\left(\left|\mathbb{E}\left[Y_N \mid I\right] - \mathbb{E}\left[g_N\left(z\right) \mid I\right]\right| \geq q_{N,\alpha/2}\right) + \mathbb{P}\left(\left|\mathbb{E}\left[Y_T \mid I\right] - \mathbb{E}\left[g_T\left(z\right) \mid I\right]\right| \geq q_{T,\alpha/2}\right) \\
&\geq \mathbb{P}\left(\left|\mathbb{E}\left[Y_N \mid I\right] - \mathbb{E}\left[g_N\left(z\right) \mid I\right]\right| \geq q_{N,\alpha/2} \vee \left|\mathbb{E}\left[Y_T \mid I\right] - \mathbb{E}\left[g_T\left(z\right) \mid I\right]\right| \geq q_{T,\alpha/2}\right) \\
&= \mathbb{P}\Big(\mathbb{E}\left[Y_N \mid I\right] \notin \left[\mathbb{E}\left[g_N\left(z\right) \mid I\right] - q_{N,\alpha}, \mathbb{E}\left[g_N\left(z\right) \mid I\right] + q_{N,\alpha}\right] \\
&\qquad \vee \mathbb{E}\left[Y_T \mid I\right] \notin \left[\mathbb{E}\left[g_T\left(z\right) \mid I\right] - q_{T,\alpha}, \mathbb{E}\left[g_T\left(z\right) \mid I\right] + q_{T,\alpha}\right]\Big) \\
&= \mathbb{P}\Big(\mathbb{E}\left[Y_N \mid I\right] \notin \left[\mathbb{E}\left[g_N\left(z\right) \mid I\right] - q_{N,\alpha}, \mathbb{E}\left[g_N\left(z\right) \mid I\right] + q_{N,\alpha}\right] \\
&\qquad \vee \frac{1}{\mathbb{E}\left[Y_T \mid I\right]} \notin \left[\frac{1}{\mathbb{E}\left[g_T\left(z\right) \mid I\right] + q_{T,\alpha}}, \frac{1}{\mathbb{E}\left[g_T\left(z\right) \mid I\right] - q_{T,\alpha}}\right]\Big) \\
&\geq \mathbb{P}\left(\frac{\mathbb{E}\left[Y_N \mid I\right]}{\mathbb{E}\left[Y_T \mid I\right]} \notin \left[\frac{\mathbb{E}\left[g_N\left(z\right) \mid I\right] - q_{N,\alpha}}{\mathbb{E}\left[g_T\left(z\right) \mid I\right] + q_{T,\alpha}}, \frac{\mathbb{E}\left[g_N\left(z\right) \mid I\right] + q_{N,\alpha}}{\mathbb{E}\left[g_T\left(z\right) \mid I\right] - q_{T,\alpha}}\right]\right).
\end{aligned}
\tag{23}
$$

It follows that for confidence level $1 - \alpha$ that

$$
\frac{\mathbb{E}\left[Y_N \mid I\right]}{\mathbb{E}\left[Y_T \mid I\right]} \in \left[\frac{\mathbb{E}\left[g_N\left(z\right) \mid I\right] - q_{N,\alpha}}{\mathbb{E}\left[g_T\left(z\right) \mid I\right] + q_{T,\alpha}}, \frac{\mathbb{E}\left[g_N\left(z\right) \mid I\right] + q_{N,\alpha}}{\mathbb{E}\left[g_T\left(z\right) \mid I\right] - q_{T,\alpha}}\right]
\tag{24}
$$

Given the previous equation, it further holds that

$$
\begin{aligned}
\delta + \alpha &\geq \\
&\geq \mathbb{P}\left(\frac{\mathbb{E}\left[Y_N \mid I\right]}{\mathbb{E}\left[Y_T \mid I\right]} \notin \left[\frac{\mathbb{E}\left[g_N\left(z\right) \mid I\right]}{\mathbb{E}\left[g_T\left(z\right) \mid I\right]} - \epsilon_{l,\delta}, \frac{\mathbb{E}\left[g_N\left(z\right) \mid I\right]}{\mathbb{E}\left[g_T\left(z\right) \mid I\right]} + \epsilon_{u,\delta}\right]\right) \\
&\quad + \mathbb{P}\left(\frac{\mathbb{E}\left[g_N\left(z\right) \mid I\right]}{\mathbb{E}\left[g_T\left(z\right) \mid I\right]} \notin \left[\frac{\sum_i g_N\left(z_{i,I}\right)}{\sum_i g_T\left(z_{i,I}\right)} - \beta_{r,\alpha}, \frac{\sum_i g_N\left(z_{i,I}\right)}{\sum_i g_T\left(z_{i,I}\right)} + \beta_{r,\alpha}\right]\right) \\
&\geq \mathbb{P}\Big(\frac{\mathbb{E}\left[Y_N \mid I\right]}{\mathbb{E}\left[Y_T \mid I\right]} \notin \left[\frac{\mathbb{E}\left[g_N\left(z\right) \mid I\right]}{\mathbb{E}\left[g_T\left(z\right) \mid I\right]} - \epsilon_{l,\delta}, \frac{\mathbb{E}\left[g_N\left(z\right) \mid I\right]}{\mathbb{E}\left[g_T\left(z\right) \mid I\right]} + \epsilon_{u,\delta}\right] \\
&\qquad \vee \frac{\mathbb{E}\left[g_N\left(z\right) \mid I\right]}{\mathbb{E}\left[g_T\left(z\right) \mid I\right]} \notin \left[\frac{\sum_i g_N\left(z_{i,I}\right)}{\sum_i g_T\left(z_{i,I}\right)} - \beta_{r,\alpha}, \frac{\sum_i g_N\left(z_{i,I}\right)}{\sum_i g_T\left(z_{i,I}\right)} + \beta_{r,\alpha}\right]\Big) \\
&\geq \mathbb{P}\left(\frac{\mathbb{E}\left[Y_N \mid I\right]}{\mathbb{E}\left[Y_T \mid I\right]} \notin \left[\frac{\sum_i g_N\left(z_{i,I}\right)}{\sum_i g_T\left(z_{i,I}\right)} - \epsilon_{l,\delta} - \beta_{r,\alpha}, \frac{\sum_i g_N\left(z_{i,I}\right)}{\sum_i g_T\left(z_{i,I}\right)} + \epsilon_{u,\delta} + \beta_{r,\alpha}\right]\right).
\end{aligned}
\tag{25}
$$

From this follows that with at least probability $1 - \alpha - \delta$ that

$$
\frac{\mathbb{E}\left[Y_N \mid I\right]}{\mathbb{E}\left[Y_T \mid I\right]} \in \left[\frac{\sum_i g_N\left(z_{i,I}\right)}{\sum_i g_T\left(z_{i,I}\right)} - \epsilon_{l,\delta} - \beta_{r,\alpha}, \frac{\sum_i g_N\left(z_{i,I}\right)}{\sum_i g_T\left(z_{i,I}\right)} + \epsilon_{u,\delta} + \beta_{r,\alpha}\right].
\tag{26}
$$

### B.3 DEBIASED RATIO ESTIMATION

The naive ratio estimator is biased due to the limited number of samples. Here we extend Popordanoska et al. (2022) to derive a debiased ratio estimator to $\mathcal{O}(n^{-2})$. Firstly, the naive estimator is:

$$
\hat{r} = \frac{\bar{y}}{\bar{x}} = \frac{\mu_y}{\mu_x}\left(\frac{\bar{y}}{\mu_y}\right)\left(\frac{\bar{x}}{\mu_x}\right)^{-1} = \frac{\mu_y}{\mu_x}\left(1 + \frac{\bar{y} - \mu_y}{\mu_y}\right)\left(1 + \frac{\bar{x} - \mu_x}{\mu_x}\right)^{-1}.
\tag{27}
$$

Then we expand $\left(1 + \frac{\bar{x} - \mu_x}{\mu_x}\right)^{-1}$ in Taylor series:

$$\hat{r} = \frac{\mu_y}{\mu_x}\left(1 + \frac{(\bar{y} - \mu_y)}{\mu_y} - \frac{(\bar{x} - \mu_x)}{\mu_x} - \frac{(\bar{x} - \mu_x)(\bar{y} - \mu_y)}{\mu_y\mu_x} + \frac{(\bar{x} - \mu_x)^2}{\mu_x^2}\right.$$

$$\left. + \frac{(\bar{x} - \mu_x)^2(\bar{y} - \mu_y)}{\mu_x^2\mu_y} - \frac{(\bar{x} - \mu_x)^3}{\mu_x^3} - \frac{(\bar{x} - \mu_x)^3(\bar{y} - \mu_y)}{\mu_x^3\mu_y} + \frac{(\bar{x} - \mu_x)^4}{\mu_x^4}\right) + \mathcal{O}(n^{-2.5})$$

(28)

The bias of $\hat{r}$ defined by $\mathbb{E}[\hat{r}] - r$ is written as:

$$\text{Bias}_r = \frac{\mu_y}{\mu_x}\left(\frac{1}{n}\left(\frac{\text{Var}(x)}{\mu_x^2} - \frac{\text{Cov}(x,y)}{\mu_x\mu_y}\right) + \frac{1}{n^2}\left(\frac{(\text{Cov}(x^2, y) - 2\mu_x\text{Cov}(x,y))}{\mu_x^2\mu_y}\right.\right.$$

(29)

$$\left.\left. - \frac{(\text{Cov}(x^2, x) - 2\mu_x\text{Var}(x))}{\mu_x^3} - \frac{3\text{Var}(x)\text{Cov}(x,y)}{\mu_x^3\mu_y} + \frac{3\text{Var}(x)^2}{\mu_x^4}\right)\right)$$

(30)

And a second-order debiased estimator is defined by $r_{corr,2} := \hat{r} - \text{Bias}_r$:

$$r_{corr,2} = \hat{r} - \frac{\mu_y}{\mu_x}\left(\frac{1}{n}\left(\frac{\text{Var}(x)}{\mu_x^2} - \frac{\text{Cov}(x,y)}{\mu_x\mu_y}\right) + \frac{1}{n^2}\left(\frac{(\text{Cov}(x^2, y) - 2\mu_x\text{Cov}(x,y))}{\mu_x^2\mu_y}\right.\right.$$

(31)

$$\left.\left. - \frac{(\text{Cov}(x^2, x) - 2\mu_x\text{Var}(x))}{\mu_x^3} - \frac{3\text{Var}(x)\text{Cov}(x,y)}{\mu_x^3\mu_y} + \frac{3\text{Var}(x)^2}{\mu_x^4}\right)\right)$$

(32)

Finally, we use plug-in estimators for empirical estimation:

$$\hat{r}_{corr,2} := \frac{\widehat{\mu_y}}{\widehat{\mu_x}}\left(1 - \frac{1}{n}\left(r_b^* - r_a^*\right) - \frac{1}{n^2}\left(\frac{(\widehat{\text{Cov}(x^2, y)} - 2\widehat{\mu_x}\widehat{\text{Cov}(x,y)})}{\widehat{\mu_x^2}\widehat{\mu_y}}\right.\right.$$

$$\left.\left. - \frac{(\widehat{\text{Cov}(x^2, x)} - 2\widehat{\mu_x}\widehat{\text{Var}(x)})}{\widehat{\mu_x^3}} - \frac{3\widehat{\text{Var}(x)}\widehat{\text{Cov}(x,y)}}{\widehat{\mu_x^3}\widehat{\mu_y}} + \frac{3\widehat{\text{Var}(x)}^2}{\widehat{\mu_x^4}}\right)\right)$$

(33)

$$r_a^* = \underbrace{\frac{\widehat{\text{Cov}(x,y)}}{\widehat{\mu_x\mu_y}}}_{=r_a}\left(1 + \frac{1}{(n-1)}\left(\frac{\widehat{\mu_y}\widehat{\text{Cov}(x^2, y)} + \widehat{\mu_x}\widehat{\text{Cov}(y^2, x)}}{\widehat{\text{Cov}(x,y)}\widehat{\mu_x}\widehat{\mu_y}} - 4\right)\right.$$

(34)

$$\left. - \frac{1}{(n-1)}\left(\frac{\widehat{\text{Var}(x)}}{\widehat{\mu_x^2}} + \frac{\widehat{\text{Var}(y)}}{\widehat{\mu_y^2}} + 2\frac{\widehat{\text{Cov}(x,y)}}{\widehat{\mu_x\mu_y}}\right)\right)$$

$$r_b^* = \underbrace{\frac{\widehat{\text{Var}(x)}}{\widehat{\mu_x^2}}}_{=r_b}\left(1 + \frac{4}{(n-1)}\left(\frac{\frac{1}{2}\widehat{\text{Cov}(x^2, x)}}{\widehat{\mu_x}\widehat{\text{Var}(x)}} - 1\right) - \frac{4}{(n-1)}\frac{\widehat{\text{Var}(x)}}{\widehat{\mu_x^2}}\right).$$

(35)

## C  LLM USAGE

We use ChatGPT (OpenAI, 2025) for polishing the writing of this paper, including improving grammar and clarity. No part of the paper was generated solely by LLM. All technical content, ideas, and experimental results were created and validated by the authors.

