# OpenReview forum: "CARE: Confidence-aware Ratio Estimation for Medical Biomarkers"
_ICLR.cc/2026/Conference — ICLR 2026 Conference Withdrawn Submission_

### Official Review · Reviewer_JpZS · 2025-10-24

**Soundness:** 2
**Presentation:** 2
**Contribution:** 1
**Rating:** 2
**Confidence:** 3

**Summary:**

The paper proposes CARE (confidence-aware ratio estimation) a drop-in, model-agnostic method for uncertainty in the form of confidence intervals for segmentation biomarker ratios by decomposing into model calibration error and estimation error using Markov inequality. Experiments are conducted on several tumor segmentation datasets and the authors find that miscalibration error is dominant uncertainty source. They claim CARE is computationally efficient and that adaptive confidence intervals can provide better coverage than standard baselines such as conformal prediction.

**Strengths:**

* Proposes confidence estimation strategy for downstream ratio estimation task by decomposing uncertainty into two bounds: an estimation-based interval of $\hat{r}$ using Markov to get $\hat{r} \pm \frac{\sqrt{\hat{\text{SE}}}}{\sqrt{\alpha}}$ and calibration-based interval using calibration dataset (similar to conformal prediction) on the numerator and denominator at a segmentation instance level. The final interval is a sum of these two components.
* The proposed method has similar computational overhead to conformal prediction and does not require modification to model, only calibration data similar to conformal prediction.
* The authors validate their method on multiple datasets (MSD-Task01, BraTS21, KiTS23) and across multiple SOTA models (nnUNet, nnFormer, UNETR++)

**Weaknesses:**

* The method is designed for ratios of two volume estimates $r = \frac{\text{Vol}(A)}{\text{Vol}(B)}$ and the confidence bounds are specific to this ratio structure. This limits applicability to other biomarkers that are not ratios and makes the contribution narrower and of limited interest to broader ICLR community.
* Fig. 7 shows that miscalibration interval (ECE)  accounts for most of the overall interval, implying that the estimation-based components is unnecessary. This suggest the contribution from the estimation-based bound may add little practical value and calls into question the entire two-part uncertainty decomposition. A simpler variant using only the calibration based interval should be compared to the full CARE method.
* The uncertainty bounds may be overly conservative due to union bound and dependence between the estimation and miscalibration error.
* The experiments are unconvincing as the baselines are either flawed or unfairly compared. The resampling baselines perform unusually low (~10% coverage in Table 1) suggesting unfair or incorrect implementation. Sampling individual pixels independently is inherently flawed as it violates spatial correlation in segmentation and should not be used for comparison. Similarly, the Bayesian methods also have poor performance perhaps due to inappropriate choice of priors. For conformal prediction, the paper demonstrates that miscalibration is dominant source of uncertainty on tumor segmentation tasks but conformal prediction is applied directly to miscalibrated model outputs, which is good practice. A stronger baseline would be to first apply post-hoc calibration e.g. temperature scaling and then apply conformal prediction. Additionally there is a lack of more modern and relevant uncertainty quantification baselines. This lack of comparison to more recent state-of-the-art techniques weakens the evaluation

**Questions:**

* Could you please provide an ablation study that evaluates the performance (coverage and adaptiveness) of a simpler method using only the calibration-based interval?
* The paper identifies miscalibration as the key issue. Have you compared CARE against a stronger baseline of "Post-hoc Calibration + Conformal Prediction"? This seems like a more direct and fair comparison.
* The performance of Subsampling and Bootstrap in Table 1 is extremely low, could you please clarify the implementation and explain the poor performance? What about methods that account for spatial dependence?
* Could you discuss the relationship between the estimation error and the calibration error? Is it possible to derive a tighter, combined bound by analyzing their dependence?
* Can you compare conformal prediction methods that account for pixel correlation such as https://proceedings.mlr.press/v162/angelopoulos22a/angelopoulos22a.pdf ?
* Given the method's high specificity to ratio-based biomarkers, how do you see this framework being generalized to other common biomarker types, such as lesion counts or non-ratio volume measurements?
* Why were more modern UQ baselines, which also handle calibration, not included in your evaluation?

---

### Official Review · Reviewer_BB7t · 2025-10-30

**Soundness:** 3
**Presentation:** 3
**Contribution:** 2
**Rating:** 4
**Confidence:** 5

**Summary:**

This paper proposes CARE (Confidence-Aware Ratio Estimation), a post-hoc statistical framework for generating confidence intervals for ratio-based biomarkers (e.g., necrosis-to-tumor ratio) derived from medical image segmentation outputs.
CARE analytically combines an estimation-based uncertainty term with a calibration-based uncertainty term. The method is model-agnostic, lightweight, and applicable to any pretrained segmentation network. Experiments on multiple architectures (nnUNet, nnFormer, UNETR++) demonstrate that CARE produces more reliable and adaptive confidence intervals than common uncertainty estimation baselines.

**Strengths:**

The paper provides a clean and interpretable mathematical formulation of uncertainty estimation. The derivation of the ratio confidence interval from a Markov upper bound and a calibration correction term is logically coherent and easy to follow. This level of theoretical transparency is uncommon in medical imaging works.

CARE’s linear-time computation, single-pass inference, and absence of sampling or retraining make it far more efficient than Bayesian or Bootstrap-based uncertainty approaches. Its plug-in nature makes it attractive for clinical integration where inference time is critical.

CARE is model-agnostic and requires no architectural modification, making the approach readily reproducible and easily applicable to other domains where ratio metrics are clinically important.

**Weaknesses:**

1. Despite the neat formulation, CARE essentially repackages existing statistical concepts—Markov inequality and calibration errors—without introducing a genuinely new computational mechanism. The innovation lies primarily in combining known pieces into a domain-specific framework. For a top-tier conference, the level of originality is modest.

2. The method is primarily motivated by identifying “uncertain” cases near clinical decision thresholds. However, such borderline cases are typically subject to manual review by radiologists in real-world workflows, even without algorithmic prompting. Consequently, the paper does not clearly justify the added clinical value of CARE beyond formalizing a process that already occurs in practice.

3. While the authors claim negligible computational cost, no empirical timing data or resource usage is reported. For clinical deployment, even small delays can matter, so quantitative runtime benchmarks would make the efficiency claim more convincing.

4. Most qualitative examples focus on near-threshold samples. The study lacks detailed analysis of scenarios where the model is apparently confident but actually miscalibrated far from the threshold, which could be the most dangerous case in medical AI. Such examples would strengthen the argument that CARE improves safety beyond obvious cases.

5. The results are scattered across multiple tables and figures, each addressing coverage, interval width, adaptiveness, and tunability separately. A single summary table (or a “coverage–width efficiency” plot) consolidating the key quantitative results for CARE and all baselines would greatly enhance readability and allow a more holistic assessment of CARE’s overall advantage.

**Questions:**

1. Given that borderline cases near clinical thresholds are already manually reviewed by radiologists, could the authors elaborate on the concrete added value of CARE in practical workflows? For instance, does CARE offer benefits in non-borderline or high-confidence yet miscalibrated scenarios?

2. Can the authors provide quantitative runtime benchmarks comparing CARE with Bootstrap, Conformal methods and other methods to substantiate the claim of computational overhead?

3. Have the authors examined cases where the segmentation model is confidently wrong (far from the threshold but miscalibrated)? Such analysis could demonstrate whether CARE enhances safety in situations beyond obvious uncertainty.

4. Would the authors consider adding a consolidated summary that jointly compares coverage accuracy, interval width, adaptiveness, and computational cost across all methods? This would help readers better assess CARE’s overall advantage.

---

### Official Review · Reviewer_a8jx · 2025-10-30

**Soundness:** 2
**Presentation:** 2
**Contribution:** 3
**Rating:** 4
**Confidence:** 4

**Summary:**

The submission studies ratio estimation in medical segmentation problems, a common task in clinical applications of machine learning methods. The submission proposes to use both notions of bias and miscalibration in the base predictor to construct uncertainty intervals that provide marginal coverage of the ground-truth ratio. The decomposition of uncertainty into bias and miscalibration would allow practitioners to form a better picture of the source of uncertainty. Experiments confirm the validity of the proposed procedure, and expand on properties of efficiency and adaptiveness.

**Strengths:**

- Uncertainty estimation of ratio estimation from machine learning models is an important problem.
- The idea to use both bias and miscalibration is interesting.
- Experiments are extensive.

**Weaknesses:**

- Presentation is sometimes hand-wavy.
- Certain experimental claims could be motivated and presented more clearly.

I have a few clarifying questions and I am looking forward to discussing with the authors!

**Questions:**

**Confusion about notation**

I am not sure I follow notation conventions throughout the paper. On line 179, $\\{z_i\\}_{i = 1}^n$ are defined as "per-pixel inputs". This notation is somewhat unusual. It would be more clear to define inputs are high-dimensional vectors, i.e. $z \in [0,1]^n$? Could the authors expand on this choice?

Models are interchangeably defined as acting on individual pixels, i.e. $z_i$, or $z$. Given my prior confusion about notation, I am not sure I understand what this means. Defining the segmentation model to act on individual pixels is counterintuitive, as the segmentation depends on the entire image, not individual pixels only. Could the authors clarify this?

Similarly, I am confused about the notion of Volume Bias. As it is written, I do not follow:

- What distribution is the expectation taken over? The joint distribution of images and ground-truth segmentation masks, or individual pixels with their respective label? If it is the latter, wouldn't that kill spatial information across pixels?

- Why is this a notion of volume bias, rather than pixel bias? Shouldn't the volume bias take into consideration the entire image?

**Definition of target fraction**

In Prop. 3.6, Line 239 defines $r = \mathbb{E}[y]/\mathbb{E}[x]$. Modulo my previous confusions about notation, could the authors expand on this choice rather than $r = \mathbb{E}[y/x]$?

In particular, the latter seems more natural, i.e. the expected ratio over the joint distribution of images and their respective segmentations, rather than the ratio of the expected  volumes---which does not take into consideration the dependency between $y$ and $x$. This seems like an important choice that should be motivated.

**Questions about Prop 3.6 and 3.8**

- Equations (5) and (8) seem to provide coverage guarantees of different quantities. Could the authors clarify whether the propositions consider the same quantities, and, if so, unify notation to avoid confusion?

- In their current formulation, the statements do not seem to depend on the size of the calibration set. I assume the authors mean to take the calibrated empirical quantiles of the respective quantities in each proposition? See my minor comment below about quantiles vs calibrated empirical quantiles.

- Prop. 3.6 is in terms of $\beta_{r,\alpha}$, which needs to be estimated, and consistency is guaranteed only asymptotically. This seems to break the finite-sample validity of the statement? Could the authors clarify this point, and the role of Prop. 3.7 when implementing the procedure on finite data? As an aside, would something weaker than consistency, e.g. unbiasedness, suffice to prove the statement?

**Adaptiveness**

The argument provided in Sec. 4.2 is not totally convincing. Usually, in CP literature, we argue for shorter, more efficient intervals. This is because larger intervals trivially provide coverage. Here, an argument is made in favor of larger intervals because they are adaptive to the hardness of the problem. Adaptiveness is certainly an important property, but vis-a-vis marginal guarantees, larger interval lengths do not imply adaptiveness.

Figure 5 is more convincing. However, the comparison with standard split CP may be considered unfair, since it is well known that marginal coverage does not imply conditional coverage. To make this argument more solid, I would suggest comparing with existing group-conditional calibration methods that are designed to achieve adaptive behavior (e.g., [1]). Finally, are stratifications in Fig. 5 computed with ground-truth or prediction labels?

**Experiments**

Using $C = 0.68$ in the experiments seems very specific. Could the authors motivate this choice?

The fact that standard CP does not achieve coverage in Fig. 6 is concerning. Could the authors clarify how this figure is generated? Baseline CP should achieve coverage by construction.

**Prior works**

There exist some prior work that studies metric guided uncertainty quantification in medical imaging problems that might be worth citing [2].

Furthermore, it might be good to include references to existing works on learning to reject / defer, who share a similar motivation as the introductory example of an automatic flag when the uncertainty interval includes a clinically-relevant threshold [].

**References**

[1] Ding et al. "Class-Conditional Conformal Prediction with Many Classes", 2023.

[2] Cheung et al. "Metric-Guided Conformal Bounds for Probabilistic Image Reconstruction", 2024.

[3] Cortes et al. "Learning with Rejection", 2017.

---

**Minor comments**

- Throughout the manuscript, empirical quantiles are mentioned simply as "quantiles" (for example, Prop. 3.2). I assume the authors mean the adjusted calibrated quantile that takes into consideration the finite number of samples available? Could the authors make this precise in the manuscript?
- Line 281: "which is analogous to multiple testing" might be unclear to a first-time reader not familiar with hypothesis testing, could the authors expand on this sentence?
- Computational complexity: without a formal statement of what the algorithm is, it is not possible to verify the claim that it runs in linear time (in the number of images? or in the number of inputs?)
- Lines 308 - 309: typo in validation vs calibration dataset?
- Lines 336 - 337: what are numerous forward passes for Bayesian methods? why are they intractable?

---

### Official Review · Reviewer_owTF · 2025-11-10

**Soundness:** 2
**Presentation:** 3
**Contribution:** 3
**Rating:** 2
**Confidence:** 2

**Summary:**

The paper addresses the problem of producing uncertainty-aware confidence intervals for ratio-based biomarkers frequently used in medical image segmentation tasks. The authors propose a framework aimed at improving the reliability of these ratio estimates, particularly in clinical contexts where uncertainty quantification is critical.

**Strengths:**

- The focus on uncertainty-aware predictions is highly relevant for healthcare applications, where reliability and transparency are essential for real-world deployment.
- The experimental evaluation is well designed, encompassing two tumor segmentation tasks and comparing performance across two widely used architectures, nnUNet and nnFormer.

**Weaknesses:**

- The motivation for restricting the framework to ratio-based predictions is insufficiently justified. It remains unclear why the proposed approach could not be extended to other types of continuous or derived metrics.
- The uncertainty quantification baselines are somewhat limited, focusing only on sampling-based methods, bootstrapping, and conformal prediction. The paper would be strengthened by comparisons with more advanced or recent UQ approaches, such as Bayesian deep learning or ensemble-based methods.

**Questions:**

- Could the authors present the implementation details of the proposed method in algorithmic form to improve reproducibility and clarity?
- What are the theoretical or methodological reasons that make the proposed approach specific to ratio estimators? Please elaborate on the assumptions or constraints that prevent its generalization to other predictive quantities.
- Can you run experiments for other types of ratios and models?

---

### Note · Authors · 2025-11-12

I have read and agree with the venue's withdrawal policy on behalf of myself and my co-authors.